# Rethinking Human Intent-to-CAD: Parametric CAD Model Generation via Cooperative Multi-Task Alignment and Spatial-Aware Reinforcement Learning

**Qingwang Zhang** [1] [*]   **Jiahao Li** [1] [*]   **Xiangdong Zhou** [1]

## Abstract

Parametric Computer-Aided-Design (CAD) modeling from human intent remains challenging, particularly during the conceptual design stage, where design goals are expressed through incomplete and unstructured modalities (*e.g.*, hand-drawn sketches and textual descriptions). In this work, we rethink the human intent-to-CAD pipeline and propose a unified method that directly maps multi-level human intents to executable codes, without assuming the prior existence of target CAD models. To support our study, we construct *HiCAD*, the first large-scale dataset aligning hand-drawn sketches, textual descriptions, and parametric CAD codes. Based on this, we introduce `HiCAD`, a two-stage framework comprising Cooperative Multi-Task Alignment to bridge the representational gap between heterogeneous inputs, and Spatial-Aware Reinforcement Learning to enforce geometric and topological consistency. Extensive experiments demonstrate that our method significantly outperforms existing baselines across multiple tasks, validating its effectiveness and robustness in transforming heterogeneous human intents into high-fidelity parametric CAD models. Our project page: `https://zqwlearning.github.io/HiCAD`.

## 1. Introduction

Computer-Aided Design (CAD) modeling is a critical step in engineering design and manufacturing processes (Brière-Côté et al., 2012). However, CAD modeling typically relies on highly specialized knowledge and complex interactive operations. Designers must precisely define geometric structures, constraint relationships, and parametric rules, making

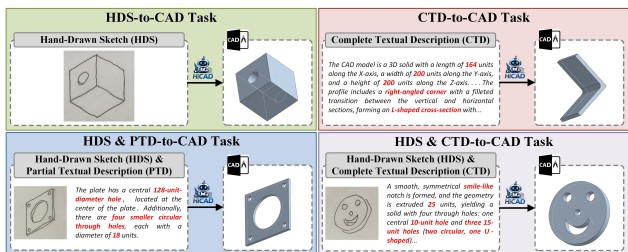

*Figure 1.* We rethink the human intent-to-CAD pipeline, aiming to explore multi-level human intent-to-CAD tasks expressed using four types of hand-drawn sketches and textual descriptions.

the prototyping process costly and time-consuming (Camba et al., 2016). With the advancement of generative models (Ho et al., 2020) and large-scale pre-trained models (Touvron et al., 2023; Bai et al., 2025b), automating or semi-automating CAD model generation has garnered increasing attention from both academia and industry (Wu et al., 2021; Li et al., 2025b; Xu et al., 2024; Li et al., 2025a).

Existing Image-to-CAD (You et al., 2025; Chen et al., 2025) and Point Cloud-to-CAD (Ma et al., 2024; Khan et al., 2024a; Rukhovich et al., 2025) pipelines are primarily based on the reconstruction paradigm. They rely on inputs like CAD model-generated renderings, photographs of physical objects, or 3D scans, all of which presuppose *the prior existence of the target CAD model in the physical or virtual domain*. However, in the early conceptual design stage, CAD models often do not exist, and the design intent is unstructured and incomplete. This generative demand—creating something from nothing—places higher requirements on current generative models (Khan et al., 2024b; Doris et al., 2025; Kolodiazhnyi et al., 2025; Li et al., 2025b).

In this paper, we rethink the human intent-to-CAD pipeline. *We posit a scenario where users convey their design goals solely through hand-drawn sketches, textual descriptions, or a combination of both.* This setup is both common and practical during the conceptual design phase: sketches capture intuitive topology and spatial layouts, while text provides high-level semantic abstractions and precise dimensional constraints. We illustrate these proposed tasks in Figure 1.

Although these modalities are highly complementary in terms of information density and geometric precision, ef-

[*]Equal contribution  [1]College of Computer Science and Artificial Intelligence, Fudan University, Shanghai, China. Correspondence to: Xiangdong Zhou <xdzhou@fudan.edu.cn>.

*Proceedings of the 43rd International Conference on Machine Learning*, Seoul, South Korea. PMLR 306, 2026. Copyright 2026 by the author(s).

fectively integrating such heterogeneous inputs and mapping them to rigorous CAD modeling logic (*e.g.*, CadQuery scripts) presents three core challenges: (i) **Asymmetric Alignment**: A substantial representational gap exists between the sparse ambiguity of hand-drawn sketches and the abstract nature of textual descriptions. (ii) **Logical Consistency**: CAD codes are highly sensitive to syntax and geometric topology; minor sequence prediction deviations can cause compilation failures or geometric distortions. (iii) **Sparse Feedback**: In complex parametric modeling spaces, traditional cross-entropy loss struggles to capture geometric fidelity in 3D space.

To overcome these challenges, we first construct the *Hi-CAD* dataset—the first large-scale benchmark featuring align hand-drawn sketches, textual descriptions, and parametric CAD code—establishing a foundation for mapping low-fidelity human intents to high-fidelity parametric CAD models. Building upon this, we propose a two-stage `HiCAD` framework that employs a unified model to handle diverse inputs and generate CAD models, as shown in Figure 1. The first stage is **Cooperative Multi-Task Alignment** (CMTA). By coordinating multiple tasks to fine-tune the foundational visual language model (VLM), CMTA achieves alignment between diverse human intents and structured CAD code, while promoting positive transfer across tasks. The second stage introduces **Spatial-Aware Reinforcement Learning** (SARL), which leverages Group Sequence Policy Optimization (Zheng et al., 2025) together with a spatial-aware reward function to enforce geometric and topological consistency. This design enables stable multi-task reinforcement learning optimization and further improves the fidelity of the generated CAD models. Our contributions can be summarized as follows:

- We rethink the human intent-to-CAD pipeline. Focusing on the conceptual design stage and creative design needs from scratch, we introduce four tasks that take combinations of hand-drawn sketches and textual descriptions as inputs to generate parametric CAD models from human intents.

- We construct a large-scale dataset aligning hand-drawn sketches, textual descriptions, and CadQuery codes, providing a unified foundation for studying the mapping from low-fidelity, unstructured human intents to parametric CAD representations.

- We propose `HiCAD`, a two-stage framework that combines Cooperative Multi-Task Alignment and Spatial-Aware Reinforcement Learning to achieve effective intent–code alignment and stable multi-task optimization with geometric and topological consistency.

- Extensive experiments demonstrate that our method outperforms existing baselines across multiple tasks, validating the effectiveness and robustness of the frame-

work in transforming multi-level human intents into parametric CAD models.

## 2. Related Work

### 2.1. Parametric CAD Sequence Modeling

The generation and reconstruction of CAD models represent core challenges in computer vision and computer-aided design. Compared to tree-based representations using Constructive Solid Geometry (Friedrich et al., 2019; Kania et al., 2020; Yu et al., 2023) or topological geometry representations like Boundary-Representation (B-Rep) (Dupont et al., 2022; Lambourne et al., 2021), modeling approaches based on parametric command sequences align more closely with the design paradigms of mainstream CAD software. This methodology enables users to construct 3D geometry through sketch-driven, incremental modeling operations. Such approaches (Wu et al., 2021; Khan et al., 2024b; Xu et al., 2022; Zhang et al.; Ma et al., 2023; Li et al., 2024; 2025b; Alrashedy et al.) generate editable, executable modeling programs and inherently align with rapidly advancing large language models, garnering increasing attention.

Recent works (Rukhovich et al., 2025; Kolodiazhnyi et al., 2025; Doris et al., 2025; Niu et al., 2025) further represent CAD command sequences as executable CadQuery (Cad-Query Developers, 2024) code. This format maintains expressive power while supporting industrial standards, serving as a crucial intermediate representation bridging intent modeling and deployable designs. This paper similarly adopts CadQuery as the CAD expression format.

### 2.2. LLM and VLM for CAD

Recent advances (Wu et al., 2023; Li et al., 2025b; Wang et al., 2025; Guan et al., 2025; Rukhovich et al., 2025; Kolodiazhnyi et al., 2025) have extended Large Language Models (LLMs) (Ouyang et al., 2022; Touvron et al., 2023; Guo et al., 2025) and Vision-Language Models (VLMs) (Liu et al., 2023; Bai et al., 2025a) to the CAD domain by treating structural geometry as a domain-specific language. Early works (Li et al., 2025b; Zhang et al.) primarily employ Supervised Fine-Tuning (SFT) on large-scale CAD datasets (Wu et al., 2021; Willis et al., 2021) to inject domain-specific knowledge, enabling models to predict modeling sequences from multi-modal inputs (*e.g.*, images, point clouds, and texts). While these methods demonstrate the feasibility of learning CAD grammar and basic geometric patterns, they heavily rely on training data distributions. Consequently, SFT-based models often struggle with geometric inconsistency and topological invalidity, as token-level objectives fail to capture global structural constraints and precise design intents.

To move beyond imitation, recent studies incorporate Re-

inforcement Learning (RL) to optimize CAD generation based on execution feedback from geometry engines. RL-CAD (Yin et al.) utilizes sketch-and-revolve sequences optimized via reconstruction quality, while CADCrafter (Chen et al., 2025) implements a code inspector as an implicit reward model and fine-tunes it using DPO (Rafailov et al., 2023). CAD-Coder (Guan et al., 2025) and ReCAD (Li et al., 2025a) use GRPO (Shao et al., 2024) to optimize the supervised fine-tuned model. Despite these efforts, existing RL-based CAD frameworks frequently suffer from high variance and training instability due to reliance on vanilla token-level importance sampling. Furthermore, the design of rewards, which is particularly important in the reinforcement learning phase, still requires further in-depth research. In this work, we propose a GSPO-based (Zheng et al., 2025) spatial-aware reinforcement learning approach that achieves more stable training and superior performance.

## 3. Building *HiCAD* Dataset

To support this study, a multimodal dataset is needed that jointly contains CadQuery codes, hand-drawn sketches, and textual descriptions. However, to our knowledge, no publicly available dataset covers all three modalities simultaneously. To address this limitation, we restructured and expanded the open-source DeepCAD (Wu et al., 2021) dataset to create *HiCAD*, a new multimodal dataset designed to support our research. Each sample in *HiCAD* includes a CadQuery code (executing it yields a CAD model), a corresponding hand-drawn style 2D sketch, and a textual description. The construction process is detailed below.

### 3.1. CAD Model Represented by CadQuery

In recent years, an increasing number of research efforts (Rukhovich et al., 2025; Kolodiazhnyi et al., 2025; Guan et al., 2025; Doris et al., 2025) have adopted CadQuery code as an intermediate representation for CAD models, with its advantages primarily manifested in the following two aspects: (i) CadQuery code possesses strong modularity and interpretability, enabling full utilization of large language models' capabilities in code generation and comprehension. (ii) Multiple high-quality CadQuery datasets (Rukhovich et al., 2025; Kolodiazhnyi et al., 2025; Doris et al., 2025) are publicly available, providing a robust data foundation for related studies. Leveraging these advantages, we adopted CadQuery as the unified representation format for CAD models in the *HiCAD* dataset. We chose to use the CadQuery codes created for the DeepCAD dataset by the CAD-Recode project (Rukhovich et al., 2025).

### 3.2. Hand-Drawn Sketch

As shown in Figure 2, the workflow starts from CAD models and produces two types of hand-drawn sketches: synthe-

sized sketches and manually drawn sketches.

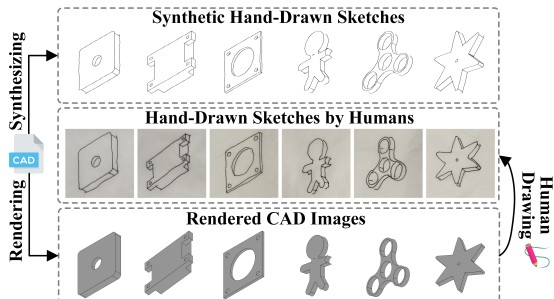

Figure 2. Illustration of the creation process for synthetic and real hand-drawn sketches.

**Synthetic Hand-Drawn Sketch.** Manually drawing hand-drawn sketches for all CAD models is impractical, so we designed a synthetic hand-drawn sketch workflow to generate a corresponding hand-drawn style sketch for each CAD model (see the **Appendix** for details). It should be noted that, during the synthesis of hand-drawn sketches, hidden lines from the CAD models are not rendered. This design decision—retaining only visible contours—aims to reduce the drawing burden on users and more accurately replicate their real-world sketching behavior.

**Hand-Drawn Sketch by Human.** Furthermore, to evaluate the generalization ability of different methods under real human sketch input, we randomly select 100 CAD models from the test set, render them as images, and then invite volunteers to draw hand-drawn sketches.

### 3.3. Textual Description

Although existing Text2CAD (Khan et al., 2024b) has provided multi-level textual descriptions for the DeepCAD dataset, its annotation workflow takes CAD command sequences (in JSON format) as inputs. We believe that text descriptions generated in this manner are not suitable as direct inputs for generating CadQuery codes, as they are inherently designed for generating command sequences. To address this, we designed a new textual description generation pipeline specifically for tasks that aim to generate CadQuery codes. The overall pipeline is shown in Figure 3.

Within this pipeline, the input information for each CAD model includes: (i) The corresponding CadQuery code; (ii) A 2D rendered image of the model; (iii) Basic geometric attribute parameters of the CAD model (*e.g.*, length, width, height, number of through-holes). For the calculation method, please refer to (Usama et al., 2025). We then inject this information into our carefully designed prompt template. See the **Appendix** for the complete prompt template. Finally, we prompt the Qwen3-VL-32B (Bai et al., 2025a) to generate the **Complete Textual Description**. This process produces descriptions for CAD models, covering

*Figure 3.* **Textual description generation pipeline.** First, we execute CadQuery code to obtain a B-rep of the CAD model for rendering the image, and a Mesh representation of the CAD model for calculating geometric attributes (*e.g.*, length, width, and height). Then, we inject the original CadQuery code, the rendered image, and the geometric attributes together into a carefully designed prompt template, which is subsequently provided to a VLM to generate a complete textual description.

only geometric shapes, structural composition, and dimensional relationships—excluding non-geometric factors such as color and material. This ensures the descriptions remain structured and consistent. Our prompt explicitly defines the model's role and output format, requiring descriptions to conform to human language conventions.

**Partial Textual Description.** After obtaining the complete textual descriptions ($\{T_{\text{CTD}}^i\}_{i=1}^N$, where $N$ is the number of samples) for all CAD models, we assume that a sample $T_{\text{CTD}}^i$ consists of $M^i$ sentences, each treated as an independent semantic unit. We then construct partial textual description ($T_{\text{PTD}}^i$) by randomly removing $k_i$ sentences from $\{T_{\text{CTD}}^i\}_{i=1}^N$. Formally, the process is defined as:

$$\mathcal{R}^i = \text{RandomSample}\big(\{1, \ldots, M^i\}, k_i\big), \qquad (1)$$

$$T_{\text{PTD}}^i = \text{Concat}\left(\big\{s_j^i \mid j \in \{1, \ldots, M^i\} \setminus \mathcal{R}^i\big\}\right). \quad (2)$$

Where $s_j^i$ denotes the $j$-th sentence in the complete textual description $T_{\text{CTD}}^i$, and $M^i$ denotes the number of sentences in $T_{\text{CTD}}^i$. The variable $k_i \sim \mathcal{U}(\{1, \ldots, M^i - 1\})$ denotes the number of sentences to be randomly removed. $\mathcal{R}^i$ denotes the index set of the removed sentences, which is obtained by uniformly sampling $k_i$ indices from $\{1, \ldots, M^i\}$. $\text{RandomSample}(\cdot, k_i)$ denotes uniform random sampling without replacement. The partial textual description $T_{\text{PTD}}^i$ is then obtained by concatenating the remaining sentences in their original order. These partial textual descriptions ($\{T_{\text{PTD}}^i\}_{i=1}^N$) are designed to simulate real-world scenarios where users provide incomplete input or only control specific aspects of the design, thereby enhancing the practical value of our research.

### 3.4. Dataset Integration and Splitting

After the above processing, a data sample contains: a hand-drawn sketch $I$, a partial textual description $T_{\text{PTD}}$, a complete textual description $T_{\text{CTD}}$, and CadQuery code $C$. As described in previous work (Xu et al., 2022), the DeepCAD dataset contains some duplicate data. To ensure data diversity and quality, we utilize DINOv3 (Siméoni et al., 2025) feature embeddings to calculate pairwise similarity between hand-drawn sketches. Samples with a similarity score exceeding 0.99 are identified as duplicates and removed.

For a fair comparison with prior works, we adopt the same data splitting as the DeepCAD dataset. As a result, in our *HiCAD* dataset, 82,075 samples are used for training, 4,536 samples for validation, and 4,015 samples for testing. It serves as a valuable resource for the CAD community and effectively facilitates research on CAD model generation from human intent.

## 4. Methodology

We propose the HiCAD framework, which generates parametric CAD models based on human intents. Next, we first formalize the problem and then introduce our framework.

### 4.1. Problem Formulation

We investigate the task of translating human intent into parametric CAD models. Essentially, it involves recognizing and parsing human intent from multi-modal user input $q$ to generate CAD models. We represent CAD models as Python scripts based on CadQuery, which generate either parametric B-Rep or 3D mesh representations when executed. Specifically, given input $q$ ($q \in \{I, T_{\text{CTD}}, (I, T_{\text{PTD}}), (I, T_{\text{CTD}})\}$, it will be explained in Section 4.2.1), we seek a trainable policy model $\pi_\theta$ such that $\pi_\theta(q) \to C$ ($C$ is essentially a Python program text used to generate the CAD model).

### 4.2. HiCAD Framework

Figure 4 shows details inside the Human intent to CAD (HiCAD) framework, which consists of two key stages: a Cooperative Multi-Task Alignment (CMTA) and a Spatial-Aware Reinforcement Learning (SARL). Our framework can accept a hand-drawn sketch, a textual description, or a combination of both as input and output CadQuery code, which can be executed to generate the corresponding CAD model. Next, we will detail the two-stage training process.

#### 4.2.1. COOPERATIVE MULTI-TASK ALIGNMENT

To bridge diverse human design intents and parametric CAD modeling, we propose the Cooperative Multi-Task Alignment (CMTA). As the foundational stage of the HiCAD framework, CMTA initializes from a pre-trained VLM to inherit its cross-modal reasoning and code generation capa-

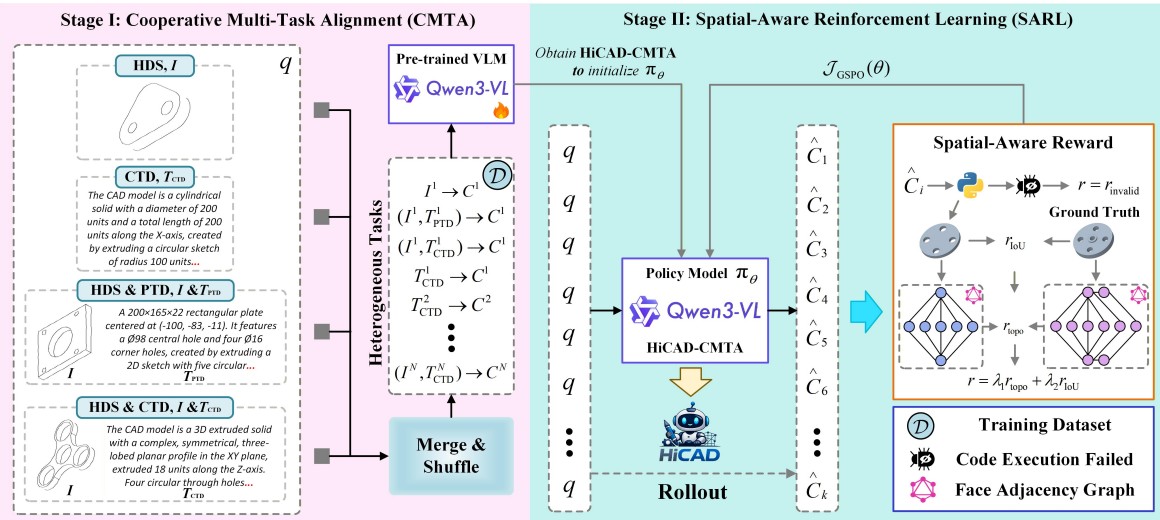

*Figure 4.* An overview of the proposed `HiCAD` framework, consisting of **Stage I: Cooperative Multi-Task Alignment (CMTA)** and **Stage II: Spatial-Aware Reinforcement Learning (SARL)**. In the CMTA stage, a pre-trained VLM is collaboratively aligned across multiple tasks. In the SARL stage, the `HiCAD`-CMTA model obtained from CMTA is used to initialize the policy model $\pi_\theta$; multiple rollouts are performed for a query $q$, and a spatial-aware reward is computed to update $\pi_\theta$ via GSPO.

bilities. We study four different forms of input to simulate human intents with varying levels of abstraction and completeness, which give rise to four corresponding tasks:

- *From Hand-Drawn Sketch to CAD, $I \to C$:* This task focuses on **geometric reconstruction**, requiring the model to infer global topology and structural relations from noisy, low-fidelity hand-drawn sketches.
- *From Complete Textual Description to CAD, $T_{\mathrm{CTD}} \to C$:* This task emphasizes **semantic alignment**, mapping abstract textual descriptions with shape and constraint information into precise geometric instructions.
- *From Hand-Drawn Sketch & Partial Textual Description to CAD, $(I, T_{\mathrm{PTD}}) \to C$:* Inspired by early-stage design, this task evaluates **multimodal completion** by integrating sketches with sparse textual cues to recover missing geometric details.
- *From Hand-Drawn Sketch & Complete Textual Description to CAD, $(I, T_{\mathrm{CTD}}) \to C$:* This task targets **high-precision fusion**, combining sketch-based spatial priors and complete textual semantics to resolve cross-modal ambiguities.

These four human intent-to-CAD tasks collectively cover the expression of human design intent from unimodal to multimodal forms and from low to high information density, closely aligning with the entire process of real-world CAD conceptual design, from early ideation to precise modeling.

The CMTA training process aims to minimize the cross-entropy between the ground truth and the predicted CadQuery code tokens: $\mathbb{E}_{(q,C)\sim\mathcal{D}}[\log \pi_\theta(C|q)]$. Where $\mathcal{D}$ represents our training dataset; $q \in \{I, T_{\mathrm{CTD}}, (I, T_{\mathrm{PTD}}), (I, T_{\mathrm{CTD}})\}$ is a heterogeneous input;

$C$ represents the CadQuery code. During CMTA, we interleave data from the four tasks, which mitigates overfitting to any single task and facilitates cross-modal latent alignment, mapping diverse expressions of human intent into a unified modeling space. By leveraging task complementarity, CMTA not only improves unimodal robustness but also promotes positive performance transfer across tasks.

### 4.2.2. SPATIAL-AWARE REINFORCEMENT LEARNING

Building upon the CMTA stage, we introduce Spatial-Aware Reinforcement Learning (SARL) as a post-training stage to further bolster the model's generalization capabilities and its precision in generating complex geometric structures. While inspired by prior works (Guan et al., 2025; Kolodiazhnyi et al., 2025), our SARL departs from existing literature in several key dimensions: first, we employ the Group Sequence Policy Optimization (GSPO) (Zheng et al., 2025) algorithm to achieve superior training stability; second, we design a spatial-aware reward that jointly optimizes geometric and topological consistency; and finally, we perform reinforcement learning across multiple tasks, standing in stark contrast to previous methods that are typically confined to single-task optimization.

We treat the model obtained after the CMTA stage (`HiCAD`-CMTA) as policy $\pi_\theta$, which generates response $C$ given input query $q$. The objective of spatial-aware reinforcement learning is to maximize the sequence-level reward function $r(q, C)$ defined by the spatial-aware reward function.

**Spatial-Aware Reward Function.** During SARL, the model's understanding of CAD spatial structures is primarily guided by a carefully designed reward function. For the

CadQuery code $C$ sampled by the policy model $\pi_\theta$, we first execute it and export the corresponding CAD model. Then, we align and compare the generated model with the real model from two complementary perspectives—topological consistency and geometric consistency—to construct a comprehensive spatial-aware reward signal.

• Topological Consistency Reward. Topological consistency evaluates structural agreement between generated and ground-truth CAD models. Using the boundary representation (B-Rep), we build face adjacency graphs with OpenCascade and quantify topological differences via the graph edit distance (GED) between predicted and ground-truth graphs. The topological consistency reward is computed as:

$$r_{\text{topo}} = \frac{1}{1 + GED(G_{\text{gt}}, G_{\text{pred}})}, \quad (3)$$

where $G_{\text{gt}}$ and $G_{\text{pred}}$ denote the face adjacency graphs of the ground-truth and predicted CAD models, respectively.

This reward constrains the topological semantics of the generated CAD model, guiding it toward a structurally consistent and logically coherent construction process.

• Geometric Consistency Reward. It measures the degree of geometric overlap between predicted CAD models and ground-truth CAD models in 3D space. For multi-body CAD models, we decompose both the ground-truth and predicted models into non-intersecting submeshes and compute the total intersection volume $V_{\text{inter}}$ by summing pairwise submesh intersections. The volumes of the ground-truth and predicted models are denoted as $V_{\text{gt}}$ and $V_{\text{pred}}$, respectively, yielding the union volume $V_{\text{union}} = V_{\text{gt}} + V_{\text{pred}} - V_{\text{inter}}$. Ultimately, the Intersection over Union (IoU) is defined as the ratio of the intersection volume to the union volume: $\text{IoU} = V_{\text{inter}}/V_{\text{union}}$. Converting IoU values directly into rewards, *i.e.*, $r_{\text{IoU}} = \text{IoU}$, can provide the model with stable and continuous reward signals, effectively avoiding the problem of reward sparsity.

• Validity and Total Reward. In addition to the spatial consistency reward mentioned above, we have introduced an additional validity check to evaluate whether the generated CadQuery code $C$ can be executed correctly and successfully generate a valid CAD model. Specifically, we execute the code $C$ by invoking the Python interpreter and verify whether it can complete the CAD modeling process without triggering syntax errors or runtime exceptions. If the code is valid and yields a feasible CAD model, the final reward is defined as a linear combination of $r_{\text{topo}}$ and $r_{\text{IoU}}$. Otherwise, when it is invalid, topological and geometric rewards are omitted, and a fixed penalty ($r_{\text{invalid}}$) is assigned. Formally, the total reward is defined as:

$$r = \begin{cases} \lambda_1 r_{\text{topo}} + \lambda_2 r_{\text{IoU}}, & \text{if } \hat{C} \text{ is valid,} \\ r_{\text{invalid}}, & \text{if } \hat{C} \text{ is invalid,} \end{cases} \quad (4)$$

where $\lambda_1$ and $\lambda_2$ are weight coefficients.

**Optimization via GSPO.** During SARL training, for each input $q$, the old policy $\pi_{\theta_{\text{old}}}$ generates a group of $k$ CadQuery code candidates $\{\hat{C}_i\}_{i=1}^{k}$. Each candidate is evaluated by the proposed Spatial-Aware Reward Function $r$, and its advantage is estimated by normalizing the reward within the sampled group. We adopt Group Sequence Policy Optimization (GSPO) to update the policy. Unlike token-level policy optimization methods, GSPO computes the importance ratio at the sequence level using the length-normalized likelihood ratio between the updated policy and the behavior policy, and applies clipping to the entire generated sequence. This design aligns the granularity of reward assignment with that of policy optimization, which is particularly beneficial for long CadQuery code generation. Formally, for a group of $k$ codes $\{\hat{C}_i\}_{i=1}^{k}$ sampled from the old policy $\pi_{\theta_{\text{old}}}$ given the same input $q$, GSPO maximizes a clipped surrogate objective defined over sequence-level importance ratios:

$$\mathcal{J}_{\text{GSPO}}(\theta) = \mathbb{E}_{q\sim\mathcal{D}, \{\hat{C}_i\}_{i=1}^{k}\sim\pi_{\theta_{\text{old}}}(\cdot|q)}$$
$$\left[ \frac{1}{k}\sum_{i=1}^{k} \min(s_i(\theta)\hat{A}_i, \text{clip}(s_i(\theta), 1-\varepsilon, 1+\varepsilon)\hat{A}_i) \right], \quad (5)$$

where $\hat{A}_i$ denotes the group-normalized advantage of the $i$-th sampled code, and $s_i(\theta)$ is the length-normalized sequence-level importance ratio. $\hat{A}_i$ and $s_i(\theta)$ are computed as follows:

$$\hat{A}_i = \frac{r(q, \hat{C}_i) - \text{mean}(\{r(q, \hat{C}_i)\}_{i=1}^{k})}{\text{std}(\{r(q, \hat{C}_i)\}_{i=1}^{k})}, s_i(\theta) = \left( \frac{\pi_\theta(\hat{C}_i|q)}{\pi_{\theta_{\text{old}}}(\hat{C}_i|q)} \right)^{\frac{1}{|\hat{C}_i|}},$$
$$(6)$$

where $|\hat{C}_i|$ denotes the sequence length of the generated CadQuery code. By assigning a shared sequence-level importance weight to all tokens within the same generated code, GSPO mitigates the instability caused by highly variable token-level ratios and leads to more stable policy updates in long-form CAD code generation.

Our SARL stage is conducted under a **joint multi-task training** setting, which preserves the model's multi-task generation capability while further improving the geometric consistency, topological validity, and syntactic validity of the generated CAD codes.

## 5. Experiment

### 5.1. Experimental Setup

**Implementation Details.** For the Cooperative Multi-Task Alignment stage, we adopt Qwen3-VL-4B-Instruct (Bai et al., 2025a) as the base model. In this stage, the per-device batch size is set to 8, the learning rate is set to $1.0 \times 10^{-5}$, and a cosine learning rate scheduling strategy is employed. The training is conducted for a total of 2 epochs. For the Spatial-Aware Reinforcement Learning stage, the learning

| Method | HDS-to-CAD Task | | | | CTD-to-CAD Task | | | | HDS & PTD-to-CAD Task | | | | HDS & CTD-to-CAD Task | | | |
|---|---|---|---|---|---|---|---|---|---|---|---|---|---|---|---|---|
| | IoU (%)↑ | IR (%)↓ | Mean CD↓ | Med. CD↓ | IoU (%)↑ | IR (%)↓ | Mean CD↓ | Med. CD↓ | IoU (%)↑ | IR (%)↓ | Mean CD↓ | Med. CD↓ | IoU (%)↑ | IR (%)↓ | Mean CD↓ | Med. CD↓ |
| GPT-4o | 20.49 | 7.00 | 53.52 | 35.31 | 47.64 | 20.80 | 25.86 | 5.93 | 37.41 | 16.60 | 38.41 | 16.93 | 51.52 | 17.20 | 22.57 | 4.50 |
| GPT-5-mini | 9.51 | 16.00 | 67.53 | 53.23 | 53.61 | 23.60 | 21.87 | 3.98 | 38.20 | 9.80 | 33.39 | 13.08 | 52.96 | 14.40 | 19.58 | 3.42 |
| Gemini-2.5 | 23.72 | 19.60 | 44.70 | 23.51 | 52.94 | 35.80 | 28.22 | 4.95 | 40.87 | 28.20 | 34.85 | 11.48 | 54.01 | 23.20 | 26.09 | 3.27 |
| Claude-3.7 | 19.48 | 15.80 | 49.51 | 31.70 | 44.21 | 28.60 | 30.11 | 6.74 | 34.11 | 22.20 | 38.86 | 18.68 | 46.81 | 26.20 | 28.99 | 6.91 |
| Qwen-VL-Max | 22.21 | 14.20 | 44.58 | 23.17 | 46.46 | 54.20 | 28.83 | 6.21 | 36.86 | 47.20 | 35.06 | 16.59 | 52.73 | 53.20 | 23.77 | 4.48 |
| CAD-Coder† | 48.89 | 3.96 | 10.56 | 2.09 | 60.08 | 3.46 | 10.90 | 0.70 | 56.38 | 4.11 | 9.47 | 1.32 | 64.61 | 3.26 | 7.60 | 0.42 |
| Cadrille† | 66.54 | 9.41 | 7.03 | 0.57 | 67.19 | 6.82 | 12.27 | 0.43 | 66.86 | 8.12 | 9.41 | 0.58 | 72.56 | 7.57 | 7.75 | 0.30 |
| **HiCAD (Ours)†** | **69.49** | **0.37** | **4.40** | **0.46** | **74.27** | **0.17** | **6.91** | **0.29** | **76.03** | **0.42** | **3.91** | **0.30** | **80.90** | **0.42** | **3.04** | **0.22** |

*Table 1.* Results for four multi-level human intent-to-CAD tasks on the *HiCAD* dataset. † denotes training on the *HiCAD* dataset. *Median* **CD** is denoted as *Med.* **CD**. CD is multiplied by $10^3$. **Bold** and underline indicate the best and the second best result.

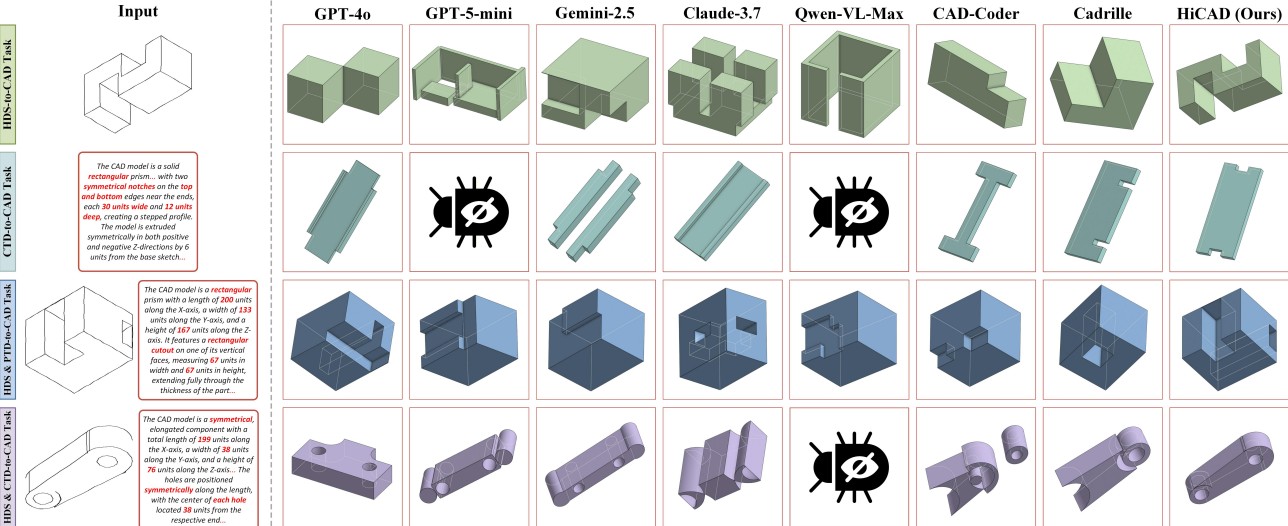

*Figure 5.* Qualitative comparison of different methods across four human intent-to-CAD tasks. ⊘ refers to an invalid and invisible model.

rate is reduced to $1.0 \times 10^{-6}$. In Equation 4, $r_{\text{invalid}}$ is set to $-0.1$, and the weighting coefficients $\lambda_1$ and $\lambda_2$ are empirically set to $0.2$ and $0.8$, respectively. Training acceleration is achieved using DeepSpeed (Rasley et al., 2020) and Flash-Attention (Dao et al., 2022). SARL is built upon the EasyR1 (Yaowei Zheng, 2025) framework, where we sample 10 candidate solutions per query $q$ during rollout, with all experiments run on 4 NVIDIA A100 80GB GPUs.

**Evaluation Metrics.** Following prior works (Rukhovich et al., 2025; Kolodiazhnyi et al., 2025; Doris et al., 2025), we evaluate the geometric accuracy between the predicted CAD models and the ground-truth CAD models using Intersection over Union (**IoU**) and Chamfer Distance (**CD**, including *Mean* and *Median* **CD**). In addition, the Invalid Rate (**IR**) is used to measure the proportion of invalid CAD models. For fair comparison, we use the Cadrille (Kolodiazhnyi et al., 2025) open-source evaluation protocol, under which all methods are evaluated.

**Baselines.** To comprehensively assess the superiority of HiCAD, we compare it against two representative categories of models: general-purpose large VLMs and CAD-domain-specific models. These two categories respectively represent cross-domain general intelligence systems and specialized optimization approaches tailored for CAD tasks. (i) **General-purpose Large VLMs**: GPT-4o (Hurst et al., 2024), GPT-5-mini (Singh et al., 2025), Gemini-2.5 (Team

et al., 2024), Claude-3.7 (Anthropic, 2025), and Qwen-VL-Max (Bai et al., 2023). These models exhibit strong joint vision–language understanding and generation capabilities, enabling them to perform image captioning, semantic alignment, and program generation in zero-shot or few-shot settings. Since these models are not trained on our dataset, we evaluate them in a 2-shot setting to provide task-specific context. (ii) **CAD-domain-specific Models**: CAD-Coder (Doris et al., 2025) and Cadrille (Kolodiazhnyi et al., 2025). These two models are state-of-the-art approaches specifically designed for parametric CAD model generation. For fair comparison, we train them on the *HiCAD* dataset based on their publicly available implementations. Other CAD-specific models (Li et al., 2025a; Rukhovich et al., 2025; Khan et al., 2024b; Niu et al., 2025; Alrashedy et al., 2024; Wang et al.) are excluded due to unavailable code or incompatibility with our multi-task setting.

### 5.2. Performance Comparison

We present a comprehensive quantitative evaluation in Table 1, comparing HiCAD against state-of-the-art specialized CAD methods (CAD-Coder, Cadrille) and leading general-purpose VLMs. The evaluation covered four tasks, representing different human intents: HDS-to-CAD, CTD-to-CAD, HDS & PTD-to-CAD, and HDS & CTD-to-CAD.

**Superiority over State-of-the-Arts.** As evidenced by the

| Method | HDS-to-CAD Task | | | | CTD-to-CAD Task | | | | HDS & PTD-to-CAD Task | | | | HDS & CTD-to-CAD Task | | | |
|---|---|---|---|---|---|---|---|---|---|---|---|---|---|---|---|---|
| | IoU (%) ↑ | IR (%) ↓ | Mean CD ↓ | Med. CD ↓ | IoU (%) ↑ | IR (%) ↓ | Mean CD ↓ | Med. CD ↓ | IoU (%) ↑ | IR (%) ↓ | Mean CD ↓ | Med. CD ↓ | IoU (%) ↑ | IR (%) ↓ | Mean CD ↓ | Med. CD ↓ |
| ST-SFT (Vanilla) | 63.84 | 1.27 | 7.68 | 0.66 | 71.01 | 1.49 | 8.91 | 0.31 | 66.61 | 1.44 | 8.07 | 0.54 | 74.35 | 1.97 | 6.53 | 0.27 |
| CMTA | 66.87 | 1.20 | 5.69 | 0.48 | 71.90 | 0.95 | 8.08 | 0.29 | 72.39 | 1.07 | 5.43 | 0.33 | 79.28 | 1.10 | 4.05 | 0.22 |
| CMTA + SARL (Ours) | 69.09 | 0.22 | 4.57 | 0.46 | 73.79 | 0.30 | 7.26 | 0.30 | 76.03 | 0.42 | 3.91 | 0.30 | 80.90 | 0.42 | 3.04 | 0.22 |

*Table 2.* Ablation study on the effectiveness of CMTA and SARL. CMTA and SARL represent our Cooperative Multi-Task Alignment and Spatial-Aware Reinforcement Learning, respectively. Notations are the same as Table 1.

results, `HiCAD` consistently outperforms all baselines across every metric and task setting. General-purpose VLMs (*e.g.*, GPT-4o, Gemini-2.5), while possessing broad semantic knowledge, struggle significantly with the strict parametric constraints of CAD modeling. They exhibit high **CD** and low **IoU** scores, indicating a lack of domain-specific geometric reasoning. Among specialized methods, while Cadrille shows competitive performance in single-modal tasks, it suffers from a high **IR**, often generating syntactically broken code. In contrast, `HiCAD` not only achieves the highest **IoU** but also ensures robust code validity, reducing **IR** by an order of magnitude (*e.g.*, from 9.41% to 0.37% on HDS-to-CAD task) compared to the strongest baseline.

**Multi-Task Alignment.** `HiCAD` demonstrates strong unified modeling and stable generalization across tasks with varying information densities and modality combinations. As shown in Table 1, its performance consistently improves as inputs expand from single-modality to multimodal settings, indicating effective fusion of complementary sketch–text information without the instability of naive concatenation or single-task training. In contrast, existing methods exhibit notable performance fluctuations under cross-task settings: general-purpose VLMs suffer accuracy degradation and increased **IR** in multimodal scenarios, while CAD-specific methods fail to maintain stability across modality changes.

Overall, `HiCAD` simultaneously enhances performance and stability in multi-level, multimodal human intent-to-CAD scenarios through multi-task collaborative alignment and unified modeling, demonstrating strong generalizability and practical application value.

**Qualitative Analysis.** Figure 5 shows the visualization results of different methods on various tasks. General VLMs often produce "hallucinated" geometries that violate basic topological rules. Specialized baselines like Cadrille, while structurally coherent, frequently fail to capture fine-grained details or complex boolean operations (*e.g.*, specific cut depths or symmetry). `HiCAD` generates CAD models that are not only geometrically precise but also topologically faithful to the user's intent, successfully reconstructing intricate features that are missed by other methods.

**Generalization to Real Human Inputs.** We evaluate all methods on a real human-drawn sketch dataset collected from volunteers (Section 3.2), with results reported in Table 3. General-purpose VLMs perform poorly, achieving **IoU** below 10%, highlighting their limited ability to handle highly ambiguous human sketches. The specialized baselines exhibit a significant domain bias between synthetic and real-world sketches. In contrast, our method exhibits substantially stronger robustness, attaining the best **IoU** while maintaining a low invalid rate. Moreover, incorporating textual descriptions effectively reduces sketch ambiguity and consistently improves performance. These results demonstrate that our approach better generalizes to real human inputs and more reliably translates imperfect sketches into accurate CAD codes, supporting its practical applicability.

| Method | HDS-to-CAD Task | | HDS & CTD-to-CAD Task | |
|---|---|---|---|---|
| | IoU (%) ↑ | IR (%) ↓ | IoU (%) ↑ | IR (%) ↓ |
| GPT-4o | 7.93 | 20.00 | 18.78 | 10.00 |
| Gemini-2.5 | 9.52 | 36.00 | 20.28 | 42.00 |
| Claude-3.7 | 8.10 | 66.00 | 23.71 | 56.00 |
| CAD-Coder[†] | 35.72 | 8.00 | 54.57 | 3.00 |
| Cadrille[†] | 17.74 | 87.00 | 59.20 | 9.00 |
| **HiCAD (Ours)**[†] | **43.27** | **3.00** | **73.37** | **0.00** |

*Table 3.* Results of different methods on HDS-to-CAD and HDS & CTD-to-CAD tasks based on real sketches drawn by humans.

### 5.3. Ablation Study

To evaluate the contribution of each core component in `HiCAD`, we perform an ablation study comparing three configurations: (i) ST-SFT (Vanilla): independent supervised fine-tuning for each task (Single Task); (ii) CMTA: Replace ST-SFT with Cross-Modal Multi-Task Alignment stage; and (iii) CMTA + SARL: Spatial-Aware Reinforcement Learning is performed after CMTA (our `HiCAD` framework). Table 2 summarizes the results across all four task settings.

**Effectiveness of CMTA.** Replacing ST-SFT with CMTA leads to continuously improve across all tasks. Notably, in the HDS & PTD-to-CAD, CMTA improves the **IoU** from 66.61% to 72.39%. These results indicate that joint training on diverse human intent-to-CAD tasks allows the model to learn a more robust, shared representation of CAD geometry. By aligning textual semantics with visual sketches, CMTA facilitates better cross-modal generalization, enabling the model to "fill in the gaps" when faced with incomplete inputs like partial text or abstract sketches.

**Impact of SARL.** Following CMTA with SARL yields further performance improvements, particularly in robustness and geometric accuracy. Across all tasks, the **IR** decreased to approximately $0.2\% \sim 0.4\%$, while the **IoU** continued to increase. This demonstrates that by explicitly rewarding spatial accuracy and valid CAD command sequences, SARL successfully bridges the gap between token-level prediction and global geometric fidelity.

In summary, CMTA provides strong semantic and cross-task representations, while SARL further enhances spatial correctness. Their combination is crucial for achieving optimal performance in our framework.

**Necessity of SARL on Multiple Tasks.** To investigate whether the SARL stage should be applied concurrently to all tasks or specialized via independent training for each task, we conduct an ablation study by varying the task compositions during the SARL stage. The results are illustrated in Figure 6. Our experiments show that applying SARL to individual tasks yields varying degrees of performance improvement across all scenarios. However, joint SARL training across multiple tasks remains essential, as this cooperative strategy substantially outperforms the single-task SARL setting and achieves superior overall performance across all evaluation metrics.

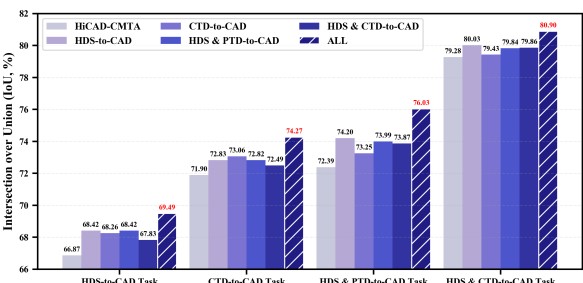

*Figure 6.* Results of training on different tasks during the SARL stage. "ALL" indicates that SARL is performed on all tasks.

## 6. Conclusion

This paper addresses the problem of generating parametric CAD models from unstructured, multimodal human intents in the conceptual design phase. By shifting the focus from reconstruction-based pipelines to intent-driven generation, we highlight the importance of jointly modeling hand-drawn sketches and textual descriptions with varying information completeness. The proposed `HiCAD` framework integrates cooperative multi-task alignment and spatial-aware reinforcement learning, enabling robust and consistent CAD code generation across multiple intent configurations. In addition, the *HiCAD* dataset provides a unified benchmark for studying human intent-to-CAD transformation under realistic design scenarios. Experimental results demonstrate that our approach achieves superior geometric fidelity and logical consistency, suggesting a promising direction for intent-centered CAD generation systems.

## Acknowledgement

We thank the reviewers' valuable suggestions. The computations in this research were performed using the CFFF platform of Fudan University.

## Impact Statement

This paper focuses on the machine learning challenge of translating human intent into parametric CAD modeling during the conceptual design phase. Our study holds promise for significantly lowering the CAD modeling barrier, enhancing prototype iteration efficiency, and advancing human-machine collaborative creation in industrial design. However, it is important to note that generated results may exhibit geometric deviations or logical errors. Practical engineering applications should incorporate human review and constraint verification to ensure reliability. Furthermore, this technology carries the potential risk of misuse in aiding the design of harmful or regulated objects.

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

# Appendix

In this Appendix, we provide the following items for a better understanding of our main paper.

## A. *HiCAD* Dataset

### A.1. Hand-Drawn Sketch

In the *HiCAD* dataset, we synthesize a hand-drawn style sketch for each CAD model. The synthesis algorithm employed is shown in Algorithm 1. Overall, the process first executes CadQuery code to export the boundary representation (B-rep) of the CAD model and stores it in STEP format. Subsequently, OpenCascade is utilized to extract the set of visible edges of the model in the nominal view. Finally, these edges are projected onto a fixed-size canvas and saved as an image file. To ensure data consistency, all synthesized hand-drawn sketches and sketches drawn by human volunteers are uniformly stored at a resolution of $512 \times 512$ pixels and used as model inputs.

---

**Algorithm 1** Synthetic Hand-Drawn Sketch Generation

---

**Require:** B-rep model $B$, image size $s$, jitter scale $\sigma$
**Ensure:** Synthetic hand-drawn sketch image $I$
1: $B \leftarrow \text{NormalizeShape}(B)$
2: $B \leftarrow \text{Triangulate}(B)$
3: $(E_v, E_h) \leftarrow \text{ComputeHLREdges}(B)$
4: $\mathcal{P}_v \leftarrow \text{ExtractPoints}(E_v)$
5: $\mathcal{P}_h \leftarrow \text{ExtractPoints}(E_h)$
6: $(\mathbf{b}_{\min}, \mathbf{b}_{\max}) \leftarrow \text{BoundingBox}(\mathcal{P}_v \cup \mathcal{P}_h)$
7: $\mathcal{P}_{img} \leftarrow \text{MapToImage}(\mathcal{P}_v, \mathbf{b}_{\min}, \mathbf{b}_{\max}, s)$
8: $I \leftarrow \text{BlankImage}(s)$
9: **for** each polyline $P \in \mathcal{P}_{img}$ **do**
10:     $P_s \leftarrow \text{Subdivide}(P)$
11:     $P_j \leftarrow \text{Jitter}(P_s, \sigma)$
12:     $I \leftarrow \text{DrawLine}(I, P_j)$
13: **end for**
14: **return** $I$

---

### A.2. Textual Description

In Figure 3 of the main paper, we present a partial prompt template employed in the text description generation process; the complete prompt template is shown in Figure 7, designed to facilitate replication of this work or further research by other scholars.

```
Instruction
You will be given a CADQuery code snippet <cadquery> and
a rendered image <image>. The CAD model has a length (X-
axis) of <length> units, width (Y-axis) of <width>
units, height (Z-axis) of <height> units, and includes
<through-holes> through holes.

Role
You are a senior CAD engineer.

Task
Generate a concise geometric description of the CAD
model. The description must follow these rules:
1. Output must be a single JSON object with one field
named "description".
2. The value of "description" must be a single paragraph
of plain English text without newline characters,
special symbols, or markdown formatting.
3. Only describe geometry and dimensions. Do not mention
color or material.
4. Combine information inferred from both the CADQuery
code and the rendered image.
5. Be written in clear, fluent, human-readable language.
6. Keep the description as concise as possible and
ensure it does not exceed 300 tokens.
7. Use "unit" for units.

Output format
{"description": "<your one-paragraph description here>"}
```

*Figure 7.* The complete prompt template used in the text description generation pipeline.

## B. Training Details

**Cooperative Multi-Task Alignment Stage.** We fine-tuned the Qwen3-VL-4B-Instruct (Bai et al., 2025a) model using Supervised Fine-Tuning (SFT). We employed a full-parameter fine-tuning approach, where the visual encoder was frozen to preserve pre-trained visual representations, while the multimodal projector and language model were updated to better align visual features with task-specific language outputs. Optimization used an Adam-based optimizer with an initial learning rate of $1 \times 10^{-5}$, combined with a cosine learning rate scheduler and a warm-up phase comprising 10% of the total training steps. bfloat16 mixed-precision training was enabled to improve computational efficiency. Training utilized DeepSpeed ZeRO-3 (Rasley et al., 2020) for memory-efficient distributed optimization.

**Spatial-Aware Reinforcement Learning Stage.** Due to computational resource constraints, we did not choose to use all the data for reinforcement training at this stage. Specifically, we selected $20k$ samples for training for each task. Hyperparameter settings in GSPO (Zheng et al., 2025) were kept at their default values.

## C. Implementation Details for Baselines

**General-Purpose VLM.** Due to computational constraints and the unavailability of source code for certain models, we are unable to fine-tune the general-purpose VLMs. To

address this, we employed a 2-shot prompting strategy to enhance its performance on CAD tasks. Specifically, we first selected 500 samples from the training dataset to construct a small knowledge base. During inference, for tasks taking hand-drawn sketches (HDS) as input—including HDS-to-CAD, HDS & PTD-to-CAD, and HDS & CTD-to-CAD—we utilize image features extracted by DINOv3 (Siméoni et al., 2025) to retrieve the two samples in the knowledge base that are semantically most similar to the input sketch. These are then constructed as contextual examples alongside the current input and fed into the model as prompts. For tasks solely input with textual descriptions (CTD-to-CAD), we employ BERT-based code embeddings (Code-BERT) (Feng, 2020) to encode the input text. Based on these embeddings, we retrieve the two text–CAD pairs most semantically similar from the knowledge base as examples, constructing corresponding 2-shot prompts.

For the aforementioned VLMs, we perform evaluation via direct API invocation. To ensure computational and financial feasibility, 500 samples are randomly selected from the test set, and the corresponding performance metrics are computed on this subset.

**CAD-Specific Models.** For specialized models targeting CAD generation tasks, we reproduced two representative approaches: CAD-Coder (Doris et al., 2025) and Cadrille (Kolodiazhnyi et al., 2025). CAD-Coder provides complete open-source code. We followed its official implementation to train the model from scratch on the *HiCAD* dataset. At the time of our experiments, Cadrille had not yet released its reinforcement learning (RL) training phase code, but the authors published the final model weights after RL fine-tuning. To ensure fair comparison, we initialized Cadrille using these RL-trained weights and continued training on the *HiCAD* dataset until convergence.

These strategies guarantee baseline models are evaluated under uniform data conditions while balancing the zero-fine-tuning constraint of general models with the best practices of specialized models.

## D. Qualitative Analysis

Figure 8 provides further qualitative comparisons of different methods across a range of tasks.

**Case Study on Real Hand-Drawn Sketches.** Figure 9 presents representative examples of our HiCAD framework on real hand-drawn sketches. The first row shows successful cases, where HiCAD generates CAD models that are largely consistent with the input sketches in terms of geometry and topology. The second row illustrates representative failure cases, in which the generated results exhibit geometric deviations or topological inconsistencies with respect to the input sketches.

## E. Limitation and Future Work

### E.1. Limitation

Despite the promising performance of HiCAD, several limitations remain:

- **Performance needs further improvement.** Although our framework performs well compared to existing methods, it still performs poorly in handling highly complex structures.

- **Ambiguity of hand-drawn sketches.** The performance of a model depends in part on the quality of the input hand-drawn sketches; highly cluttered or non-standard perspectives can sometimes lead to misaligned spatial layouts.

- **Human intent representation.** How humans express their intentions, and how models receive and understand these intentions, remains an open question. While this paper systematically introduces four tasks, this is certainly not exhaustive. We believe this remains an area for further exploration in the field of CAD.

### E.2. Future Work

- **Iterative refinement.** We plan to extend HiCAD into a conversational agent that allows users to refine the generated CAD model through multi-turn user feedback.

- **Alignment of hand-drawn sketches with textual descriptions.** Although our method demonstrates some ability to align hand-drawn sketches with text descriptions, we believe that this reliance on the model's automatic implicit alignment is insufficient, and we plan to develop explicit alignment strategies in the future.

- **Bridge the domain gap between synthetic and real hand-drawn sketches.** Although our method was trained solely on synthetic sketches, its performance on real sketches has significantly outperformed existing approaches. Nevertheless, a certain degree of performance degradation remains unavoidable. Future work will focus on developing more photorealistic sketch synthesis strategies or designing models with stronger cross-domain generalization capabilities to further narrow the domain gap between synthetic and real hand-drawn sketches.

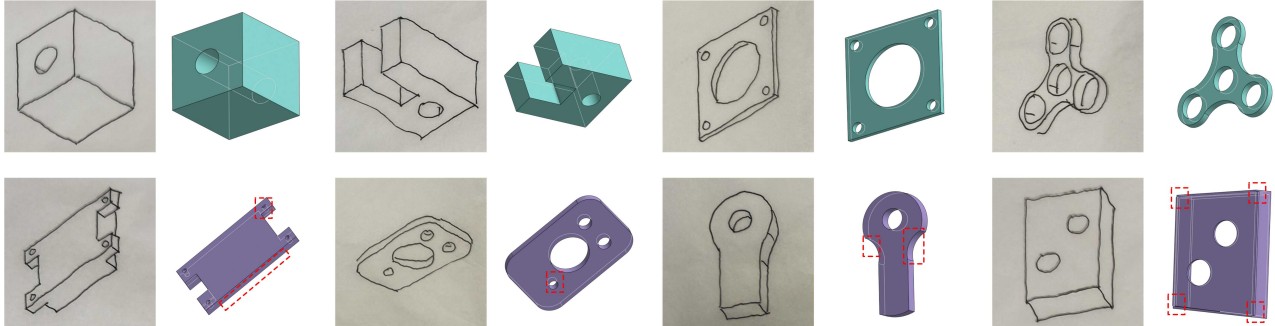

*Figure 8.* Qualitative comparison of different methods across four human intent-to-CAD tasks. ☒ refers to an invalid and invisible model.

*Figure 9.* Some examples of our `HiCAD` framework handling real hand-drawn sketches. The red dashed boxes indicate areas that are clearly incorrect.

