# OpenReview forum: "Rethinking Human Intent-to-CAD: Parametric CAD Model Generation via Cooperative Multi-Task Alignment and Spatial-Aware Reinforcement Learning"
_ICML.cc/2026/Conference — ICML 2026 regular_

### Official Review · Reviewer_ME3i · 2026-03-07

**Soundness:** 2
**Presentation:** 3
**Significance:** 3
**Originality:** 3
**Overall Recommendation:** 5
**Confidence:** 4

**Summary:**

This paper proposes HiCAD, a unified method that maps hand-drawn sketches, textual descriptions, or their combination to CAD models. The method consists of two stages. First, Cooperative Multi-Task Alignment is employed to integrate multimodal inputs. Then, Spatial-Aware Reinforcement Learning is introduced to enhance the geometric and topological consistency of the generated CAD models.

**Compliance With Llm Reviewing Policy:**

Affirmed.

**Final Justification:**

The response has addressed my concerns regarding concept definition, dataset accuracy, and methodological novelty. I will raise my score to 5.

**Key Questions For Authors:**

1. Does the concept of “human intent-to-CAD” only include scenarios where the inputs are hand-drawn sketches, textual descriptions, or their combination?

2. How to validate the quality of the synthetic dataset?

3. Please clarify the novelty of the proposed Cooperative Multi-Task Alignment.

**Limitations:**

Yes

**Strengths And Weaknesses:**

Strengths

1. The proposed spatial-aware reinforcement learning enhances the model’s ability to generate geometric structures from the perspectives of topological consistency and geometric consistency.

2. The paper is clearly organized and logically structured, making it easy to read.

Weaknesses

1.  The paper claims to “rethink the human intent-to-CAD pipeline.” However, the task formulated in this work generates CAD models from hand-drawn sketches, textual descriptions, or their combination. I question whether the description of "human intent-to-CAD pipeline" is sufficiently accurate. While I acknowledge the authors’ argument that inputs such as images or point clouds often assume the prior existence of CAD models, these modalities can also express design intent. Therefore, the corresponding CAD generation tasks should also be considered part of the “human intent-to-CAD”.

2. The proposed Cooperative Multi-Task Alignment mainly alternates the training of different task datasets. It seems relatively simple and lacks novelty.

3. In the HiCAD dataset, the sketches are generated using a synthetic algorithm, and the textual descriptions are produced by an LLM. This raises concerns about the quality of the data. Have the authors conducted validation experiments to assess the quality of the synthetic dataset?

4. The constructed HiCAD dataset includes a part called “Partial Textual Description.” In Lines 184–189, the paper claims that this data simulates scenarios where users provide incomplete inputs. However, this part of the dataset is created by randomly removing some sentences from the complete textual descriptions. I am concerned about the rationality of the "Partial Textual Description". Specifically, even when the provided information is incomplete, users’ expressions are typically coherent paragraphs. Random sentence removal may instead result in fragmented sentences, which may not align well with human expression patterns.

---

> ### Author Rebuttal · Authors · 2026-03-30
>
> Dear Reviewer ME3i,
>
> We sincerely appreciate your positive comments on the effectiveness of spatial-aware reinforcement learning (SARL) and on the clear organization, logical flow, and readability of our paper. Thank you for your valuable feedback, and we are grateful for the opportunity to clarify the points you raised.
>
> ---
>
> ## **Q1: Scope of the Human Intent-to-CAD Pipeline**
>
> Our work aims to provide a **new perspective** on the human intent-to-CAD pipeline. In this paper, we focus on the **conceptual design stage**, where low-fidelity inputs—such as hand-drawn sketches, text descriptions, or their combination—are particularly important, since a complete CAD model often does not yet exist at this stage (Lines 41–47, right column). That said, we fully acknowledge that other modalities, including **images and point clouds**, can also express design intents, and we **do not intend to exclude them** from the broader definition of human intent.
>
> More importantly, we do not claim to have fully solved the human intent-to-CAD problem, nor do we intend to restrict it to the specific tasks studied in this work. Rather, our goal is to introduce a **new research perspective** for this problem. As discussed in **Section E.1, Point 3**, the representation of human intent remains an **open research question**.
>
> ---
>
> ## **Q2: Dataset Quality**
>
> Due to the large-scale data ($ \sim{10^5}$), an automated annotation process is almost essential. Below, we discuss the quality of the **text descriptions** and **hand-drawn sketches** separately.
>
> - For the **text descriptions**, please refer to **Reviewer 8Nay, Q1**.
>
> - For the **hand-drawn sketches**, besides the algorithmically simulated sketches, we also provide a subset of **sketches drawn by humans** (**Section 3.2**). To evaluate the fidelity of the simulated sketches, we compare them with the human-drawn ones using **SSIM** [1] and **LPIPS** [2]. **SSIM** ranges from 0 to 1, with higher values indicating greater structural similarity, while **LPIPS** is non-negative, with lower values indicating higher perceptual similarity. The results show ${\rm{SSIM}} = 0.8574$ and ${\rm{LPIPS }} = {\rm{ }}0.0803$, indicating that the simulated sketches are **highly consistent** with human-drawn sketches in both structural and perceptual appearance. More examples of human-drawn sketches and their corresponding simulated sketches are provided in the **supplementary material**.
>
> In summary, we believe that our dataset provides a **large-scale, high-quality, and reliable benchmark** for studying human intent-to-CAD tasks.
>
> ---
>
> ## **Q3: Novelty of CMTA**
>
> The novelty of **CMTA** lies in its cooperative alignment of CAD tasks across both modality and information density within a shared executable CAD code space. CMTA does not merely combine multiple tasks; rather, it is deliberately designed to jointly align a set of CAD tasks spanning **unimodal to multi-modal inputs** and **low- to high-information-density settings**, enabling different tasks to complement one another within a **shared code space** and yielding significant **positive transfer effects** (Lines 253–259, left column).
>
> ---
>
> ## **Q4: Regarding the Rationality of the "Partial Textual Description"**
>
> To better evaluate whether our **partial textual description (PTD)** setting is reasonable, we further tested several partial-text inputs that more closely resemble real human expression patterns (**inspired by Reviewer 8Nay, Q2**). Specifically, we used GPT-4o to retain three common types of information from the complete textual descriptions in the test set: **geometric attribute descriptions**, **topology-related descriptions**, and **global shape descriptions**. We then directly evaluated these inputs using the trained model, with results shown below:
>
> | Input                                     | IoU (%) $ \uparrow $ |
> | ----------------------------------------- | -------------------- |
> | Hand-Drawn Sketch (HDS)                   | 69.49                |
> | HDS with geometric attribute descriptions | 80.76                |
> | HDS with topology-related descriptions    | 75.33                |
> | HDS with global shape descriptions        | 73.41                |
> | HDS with partial textual descriptions     | 76.03                |
>
> These results show that the model trained with our current **random sentence removal** strategy generalizes well to partial textual descriptions that are closer to real human inputs. We believe this is because random removal **reduces the risk of overfitting to specific omission patterns** during training. Therefore, the current PTD setting can be regarded as a **simple, effective, low-cost, and scalable approximation of real human partial-text input**.
>
> ---
>
> ### **References**
>
> [1] Wang et al., Image Quality Assessment: From Error Visibility to Structural Similarity, TIP, 2004.
>
> [2] Zhang et al., The Unreasonable Effectiveness of Deep Features as a Perceptual Metric, CVPR, 2018.

---

> > ### Author Rebuttal · Reviewer_ME3i · 2026-04-02
> >
> > Thanks for the response. These clarifications have addressed all of my concerns, and I will raise my score to 5. I still recommend that the authors incorporate the “Q1: Scope of the Human Intent-to-CAD Pipeline” section from the response into the revised paper to provide a more precise definition of the “Human Intent-to-CAD” concept.

---

> > > ### Author Response · Authors · 2026-04-03
> > >
> > > Dear Reviewer ME3i,
> > >
> > > We greatly appreciate your thoughtful feedback and are pleased to see that our response has satisfactorily resolved your concerns.
> > >
> > > We also thank you for your valuable suggestion to incorporate the "Q1: Scope of the Human Intent-to-CAD Pipeline" section into the revised manuscript. We agree that doing so will provide a more precise definition of the "Human Intent-to-CAD" concept. We will incorporate this clarification into the revised paper.
> > >
> > > Thank you again for your valuable comments and support.
> > >
> > >
> > > Best regards,

---

### Official Review · Reviewer_8Nay · 2026-03-12

**Soundness:** 2
**Presentation:** 3
**Significance:** 3
**Originality:** 3
**Overall Recommendation:** 4
**Confidence:** 4

**Summary:**

This paper studies intent-to-CAD in the conceptual design setting, where the input is not an existing object or scan but a hand-drawn sketch, a textual description, or both. The authors introduce *HiCAD*, a dataset that aligns CadQuery code with synthetic and real sketches, as well as LLM-generated textual descriptions. They also propose a two-stage framework for intent-to-CAD generation: CMTA, which jointly fine-tunes a VLM across four tasks, and SARL, which optimizes the model using a reward that combines code validity, topology via graph edit distance on face adjacency graphs, and geometry via 3D IoU.

The experiments report improvements over general-purpose VLMs and two CAD-specific baselines across four tasks, along with ablations suggesting benefits from both multi-task supervised training and the RL stage.

**Compliance With Llm Reviewing Policy:**

Affirmed.

**Key Questions For Authors:**

1. Since much of the text annotation is LLM/VLM-generated, can the authors provide more analysis of the quality and realism of these descriptions? For example, is there any human validation or error analysis?

2. For the partial-text setting, can the authors analyze which types of dropped information hurt performance the most? For example, are geometric attributes, topology-related descriptions, or global shape descriptions more important?

3. Can the authors provide more statistics or analysis on the shape distribution and diversity of *HiCAD*? This would help clarify how broad and representative the benchmark is.

4. Can the authors include stronger reward ablations to separate the contributions of the validity reward, topology reward, and IoU reward?

**Limitations:**

Yes

**Strengths And Weaknesses:**

### Strength

- Realistic problem setting: The paper tackles intent-to-CAD generation from sketch, text, or both, which is more practical for conceptual design than reconstruction from an existing shape.

- Convincing dataset construction pipeline aligned with real-world scenarios: The dataset design includes several thoughtful choices, such as not rendering hidden lines for the synthetic hand sketches, randomly dropping sentence descriptions, and applying diversity control.

- Strong experimental results: The method shows clear gains over the reported baselines across multiple tasks, and the ablations indicate that both CMTA and SARL contribute to the final performance.

### Weakness

- Dataset realism is still limited: Much of text description from HiCAD is generated by LLM/VLMs, so the “human intent” setting may be less realistic than claimed.

- Experimental analysis is limited: For the partial-text setting, the paper would benefit from an analysis of which aspects of the dropped sentence descriptions affect performance the most.

- Insufficient analysis of dataset diversity: The paper would be stronger with more analysis of the shape distribution and diversity of the proposed dataset.

- Limited mechanism analysis: The paper lacks stronger reward ablations to isolate whether the gains come from the topology reward, IoU reward, validity reward, or simply from multi-task SFT.

#### Minor Issues
- Equation 4: The definition of $s_i(\theta)$ may contain a typo. As written, the numerator and denominator appear to be the same, which would make the ratio degenerate.

---

> ### Author Rebuttal · Authors · 2026-03-30
>
> Dear Reviewer 8Nay,
>
> We greatly appreciate your recognition of our problem setting, dataset creation, and experimental results. We sincerely thank you for your valuable feedback and are pleased to address your concerns.
>
> ---
>
> ## **Q1: More Analysis of the Quality and Realism of VLM-Generated Text Descriptions**
>
> It is important to clarify that the text descriptions are **not unconstrained outputs directly generated by the VLM**. Instead, we impose **explicit rule-based constraints** during the generation process. Specifically, in the prompt design, we require the VLM to follow a set of predefined rules, such as adopting **human-like descriptive expressions** and focusing on **geometric and structural information**. The detailed prompting rules are shown in **Figure 7**, and more examples of generated text descriptions are provided in the **supplementary material**.
>
> Furthermore, to better evaluate the realism of the generated descriptions, we introduce **human-corrected text descriptions** from the CADFusion dataset [1] during the testing phase of the **HDS-to-CAD** task (HDS: Hand-Drawn Sketch). The results show a clear performance improvement, with **IoU increasing from $69.49\\%$ to $74.44\\%$**. This result suggests that the model trained on our generated descriptions can effectively leverage real human-written descriptions at test time, indicating **strong information-level consistency** between the VLM-generated text and real human descriptions.
>
> ---
>
> ## **Q2: Analysis of the Partial Text Setting**
>
> We greatly appreciate this constructive suggestion. It has helped us further strengthen the paper, and we will consider adding the following analysis in the final version.
>
> **Motivated by this question**, we conducted a more fine-grained study of the partial-text setting by using GPT-4o to remove different types of information from the complete text descriptions in the test set. Specifically, we observe that removing **geometric attribute descriptions** causes the largest performance drop (**IoU:** $77.20\\%$), removing **topology-related descriptions** results in a moderate decline (**IoU:** $79.99\\%$), and removing **global shape descriptions** has the smallest effect (**IoU:** $80.41\\%$). These findings strongly **support our intuition** (Lines 44–47, right column): sketches primarily convey spatial layout, whereas text is more effective at expressing precise geometric and dimensional constraints.
>
> For reference, the **IoU** of the **HDS & PTD-to-CAD** task is $76.03\\%$, while that of the **HDS & CTD-to-CAD** task is $80.90\\%$ (**Table 1**, PTD: Partial Textual Description, CTD: Complete Textual Description) . From this perspective, the partial-text setting implemented via random removal can be regarded as a **more challenging** yet **cost-effective** proxy for real human input, as it does not depend on either additional human annotation or large-model generation.
>
> ---
>
> ## **Q3: More Statistics of *HiCAD* Dataset**
>
> We provide additional statistics of *HiCAD* from the perspectives of **CAD command sequence length** (**2~5:** 75,347; **6~9:** 12,854; **≥10:** 2,425) and **extrusion count** (**1:** 49,521; **2~3:** 33,999; **4~5:** 5,842; **≥6:** 1,264). We will add more detailed charts and analysis in the final version.
>
> ---
>
> ## **Q4: Reward Ablation**
>
> For the contribution of each reward term, we agree that a clearer presentation is needed. As shown below, removing any reward component degrades performance. Notably, removing $r_{\rm IoU}$ causes the largest IoU drop, while removing $r_{\rm invalid}$ significantly increases IR. These results indicate that the reward terms play complementary roles in ensuring **geometric fidelity**, **topological validity**, and **executable correctness**.
>
> *From left to right: **HDS-to-CAD** and **CTD-to-CAD** tasks. The result for “**Ours**” is extracted from **Table 1**.*
>
> |           Method           | IoU (%) $ \uparrow $ | IR (%) $ \downarrow $ | IoU (%) $ \uparrow $ | IR (%) $ \downarrow $ |
> | :------------------------: | :------------------: | :-------------------: | :------------------: | :-------------------: |
> |  w/o ${r_{{\rm{topo}}}}$   |        68.67         |         0.57          |        73.30         |         0.60          |
> |   w/o ${r_{{\rm{IoU}}}}$   |        66.82         |         0.50          |        72.44         |         0.50          |
> | w/o ${r_{{\rm{invalid}}}}$ |        68.96         |         1.27          |        72.63         |         0.65          |
> |          **Ours**          |      **69.49**       |       **0.37**        |      **74.27**       |       **0.17**        |
>
> We will add this ablation study to the final version to more clearly demonstrate the role of each reward component.
>
> ---
>
> ## **Q5: Equation 4**
>
> Thank you for pointing this out. We will correct this typo in the final version.
>
> ---
>
> ### **References**
>
> [1] Wang et al., Text-to-CAD Generation Through Infusing Visual Feedback in Large Language Models, ICML, 2025.

---

> > ### Author Rebuttal · Reviewer_8Nay · 2026-04-03
> >
> > Thanks for the rebuttal. It has resolved my concern.

---

### Official Review · Reviewer_Fyp5 · 2026-03-12

**Soundness:** 3
**Presentation:** 4
**Significance:** 2
**Originality:** 3
**Overall Recommendation:** 4
**Confidence:** 4

**Summary:**

The core idea is that inputs can be either sketches or text descriptions, or both, this paper investigates human-intent-to-CAD generation. In addition to proposing a two-stage framework with multi-task alignment and spatial-aware reinforcement learning, it presents HiCAD, a multimodal dataset that aligns hand-drawn sketches, textual descriptions, and executable CadQuery code. Compared to recent CAD-specific baselines and general VLMs, experiments on four intent-to-CAD tasks demonstrate improved CAD generation quality.

**Compliance With Llm Reviewing Policy:**

Affirmed.

**Key Questions For Authors:**

1)How much of the performance gain comes specifically from the joint sketch-text training, as opposed to the RL stage? Authors provide stronger module wise and component wise ablations?

2)Could the authors clarify what they see as the main methodological novelty of the proposed approach compared to recent CAD generation works?

**Limitations:**

Yes

**Strengths And Weaknesses:**

## Strengths

1) Paper is well written and is easy to understand.
 2) Paper propose a useful multimodal dataset that aligns  sketches textual description and cad query code, supporting the research in cad generation .
3) Proposed framework is well engineered and strong empirically , showing good overall CAD specific baselines.

## Weaknesses

1) The novelty is incremental because the paper primarily integrates  components multimodal conditioning, multi task fine tuning, and RL-based CAD refinement into a unified framework for sketch/text-to-CAD generation.
2) Paper could gain more if it have a ablation study that provides evidence  of each component contribution in the gains in results.
3) The dataset seems carefully built, but it dose not extends existing assets/cad operations  and also annotation pipelines incremental rather than introducing fundamentally concept or  data .

---

> ### Author Rebuttal · Authors · 2026-03-30
>
> Dear Reviewer Fyp5,
>
> We sincerely thank you for your positive evaluation of our paper, particularly your recognition of its writing quality, the value of the multi-modal dataset, and the design of our framework. We greatly appreciate your thoughtful feedback and are grateful for the opportunity to address your concerns.
>
> ---
>
> ## **Q1: Regarding the Two-stage Ablation**
>
> Our framework consists of two core stages, **CMTA** and **SARL** (as you mentioned). The corresponding ablation results are presented in **Section 5.3**, where **Table 2** shows that both stages yield substantial performance improvements. Furthermore, **Figure 6** further demonstrates that introducing **multi-task alignment** in the SARL stage brings a clear additional gain in performance.
>
> For the ablation study on the **reward design in the SARL stage**, please refer to our response to **Reviewer 8Nay, Q4**.
>
> ---
>
> ## **Q2: Methodological Novelty Compared to Recent CAD Generation Works**
>
> The novelty of this work does not lie in a single isolated module, but in the **first systematic formulation of the human intent-to-CAD problem in the conceptual design stage**, together with a corresponding **unified dataset** and **training framework**.
>
> Specifically, our contributions and how they differ from prior work can be summarized in the following three aspects.
>
> 1. **Problem Formulation.**
>
>    Unlike recent CAD generation or reconstruction works [1–4], which typically assume that the target object is already available as an image or point cloud, we study **direct parametric CAD model generation from incomplete and unstructured human intent**, including hand-drawn sketches, complete or partial text descriptions, or their combination. This setting is rooted in the **conceptual design phase** and is therefore more consistent with real-world creative workflows that begin from scratch.
>
> 2. **Unified Dataset.**
>
>    We construct *HiCAD*, which, to the best of our knowledge, is the **first large-scale dataset** that jointly associates **hand-drawn sketches, textual descriptions, and executable CadQuery codes** in aligned triplets.
>
> 3. **Training Framework.**
>
>    We propose a two-stage framework for $\mathtt{HiCAD}$:
>
>    - **Cooperative Multi-Task Alignment (CMTA).**
>      CMTA jointly aligns four human intent-to-CAD tasks with varying abstraction levels and information densities. This design mitigates overfitting to any single task, promotes **cross-modal latent alignment**, and maps diverse forms of human intent into a shared modeling space. By exploiting task complementarity, CMTA not only improves unimodal robustness but also yields **positive transfer across tasks**.
>    - **Spatial-Aware Reinforcement Learning (SARL).**
>      Compared with prior RL-based CAD methods [2, 5], SARL differs in three major aspects (Lines 266–273, left column): it adopts **GSPO** [6] to improve training stability, introduces a **spatial-aware reward** that jointly captures **geometric and topological consistency**, and performs **multi-task reinforcement learning** rather than conventional single-task optimization. These designs make SARL both more stable and better suited to the structural requirements of CAD generation.
>
> We hope the above clarification resolves your concerns regarding the novelty of our work.
>
> ---
>
> ## **Q3: Dataset Contribution**
>
> We thank you for recognizing the effort we devoted to constructing the dataset. We acknowledge that *HiCAD* is built upon the existing DeepCAD dataset [7] (Lines 130–133, left column). However, our contribution goes well beyond simply reusing existing resources. To the best of our knowledge, *HiCAD* is the first large-scale dataset (approximately $10^5$ triplets) that integrates **hand-drawn sketches, textual descriptions, and executable CadQuery codes** within a unified setting. The hand-drawn sketches (including algorithmic simulations and human drawings) and text descriptions are created by us. We propose *HiCAD* as the first **reliable benchmark and evaluation platform** for studying the unified mapping from multi-level, low-fidelity human intent inputs to high-fidelity parametric CAD codes.
>
> ---
>
> ### **References**
>
> [1] You et al., Img2CAD: Reverse Engineering 3D CAD Models from Images through VLM-Assisted Conditional Factorization, SIGGRAPH Asia, 2025.
>
> [2] Chen et al., CADCrafter: Generating Computer-Aided Design Models from Unconstrained Images, CVPR, 2025.
>
> [3] Khan et al., CAD-SIGNet: CAD Language Inference from Point Clouds using Layer-wise Sketch Instance Guided Attention, CVPR, 2024.
>
> [4] Rukhovich et al., CAD-Recode: Reverse Engineering CAD Code from Point Clouds, ICCV, 2025.
>
> [5] Kolodiazhnyi et al., cadrille: Multi-modal CAD Reconstruction with Reinforcement Learning, ICLR, 2026.
>
> [6] Zheng et al., Group Sequence Policy Optimization, arXiv, 2025.
>
> [7] Wu et al., DeepCAD: A Deep Generative Network for Computer-Aided Design Models, ICCV, 2021.

---

> > ### Author Rebuttal · Reviewer_Fyp5 · 2026-04-03
> >
> > Thanks for the rebuttal. The points I raised have been addressed. I am keeping the score.

---

### Decision · Program_Chairs · 2026-04-30

**Decision:**

Accept (regular)

**Comment:**

The overall recommendations are 2 weak accepts and 1 accept. The reviewers agreed that (1) this paper solves a novel and practical problem, (2) the proposed framework is well engineered, and (3) this paper shows strong experimental results. They initially raised some concerns about novelty, dataset quality, and ablation study. The authors' rebuttal has fully resolved their concerns.